# Spironolactone, a Classic Potassium-Sparing Diuretic, Reduces Survivin Expression and Chemosensitizes Cancer Cells to Non-DNA-Damaging Anticancer Drugs

**DOI:** 10.3390/cancers11101550

**Published:** 2019-10-14

**Authors:** Tomomi Sanomachi, Shuhei Suzuki, Keita Togashi, Asuka Sugai, Shizuka Seino, Masashi Okada, Takashi Yoshioka, Chifumi Kitanaka, Masahiro Yamamoto

**Affiliations:** 1Department of Molecular Cancer Science, Yamagata University School of Medicine, 2-2-2 Iida-nishi, Yamagata 990-9585, Japan; t-sanomachi@med.id.yamagata-u.ac.jp (T.S.); s-suzuki@med.id.yamagata-u.ac.jp (S.S.); ke-togashi@med.id.yamagata-u.ac.jp (K.T.); m-okada@med.id.yamagata-u.ac.jp (M.O.); ckitanak@med.id.yamagata-u.ac.jp (C.K.); 2Department of Clinical Oncology, Yamagata University School of Medicine, 2-2-2 Iida-nishi, Yamagata 990-9585, Japan; ytakashi@med.id.yamagata-u.ac.jp; 3Department of Ophthalmology and Visual Sciences, Yamagata University School of Medicine, 2-2-2 Iida-nishi, Yamagata 990-9585, Japan; 4Research Institute for Promotion of Medical Sciences, Yamagata University Faculty of Medicine, 2-2-2 Iida-nishi, Yamagata 990-9585, Japan

**Keywords:** drug resistance, spironolactone, osimertinib, survivin, xenograft, cancer stem cells

## Abstract

Spironolactone, a classical diuretic drug, is used to treat tumor-associated complications in cancer patients. Spironolactone was recently reported to exert anti-cancer effects by suppressing DNA damage repair. However, it currently remains unclear whether spironolactone exerts combinational effects with non-DNA-damaging anti-cancer drugs, such as gemcitabine and epidermal growth factor receptor tyrosine kinase inhibitors (EGFR-TKIs). Using the cancer cells of lung cancer, pancreatic cancer, and glioblastoma, the combinational effects of spironolactone with gemcitabine and osimertinib, a third-generation EGFR-TKI, were examined in vitro with cell viability assays. To elucidate the underlying mechanisms, we investigated alterations induced in survivin, an anti-apoptotic protein, by spironolactone as well as the chemosensitization effects of the suppression of survivin by YM155, an inhibitor of survivin, and siRNA. We also examined the combinational effects in a mouse xenograft model. The results obtained revealed that spironolactone augmented cell death and the suppression of cell growth by gemcitabine and osimertinib. Spironolactone also reduced the expression of survivin in these cells, and the pharmacological and genetic suppression of survivin sensitized cells to gemcitabine and osimertinib. This combination also significantly suppressed tumor growth without apparent adverse effects in vivo. In conclusion, spironolactone is a safe candidate drug that exerts anti-cancer effects in combination with non-DNA-damaging drugs, such as gemcitabine and osimertinib, most likely through the suppression of survivin.

## 1. Introduction

Cancer is one of the most common causes of death [1]. The development of therapeutic strategies for patients with advanced cancer has markedly improved overall survival. However, resistance to anticancer reagents is inevitable, and the prognosis of advanced cancer remains poor. There are several potential sources of chemoresistance, including alterations to drug transporters, the suppression of apoptosis, mitochondrial alterations, the promotion of DNA damage repair, autophagy, epithelial-mesenchymal transition, and cancer stem cells (CSCs) [2,3,4,5,6]. Appropriate chemotherapeutic strategies that consider the mechanisms of resistance are necessary to cure cancer.

Non-small cell lung cancer (NSCLC), pancreatic cancer, and glioblastoma are malignancies with poor prognoses [7], and improvements in treatments for these malignancies are awaited. Gemcitabine (2′,2′-difluorodeoxycytidine, dFdC) is a pyrimidine nucleotide antimetabolite that is incorporated into DNA and terminates DNA replication. It is one of the standard chemotherapeutic reagents for solid tumors, including NSCLC and pancreatic cancer [8,9]. However, the effects of gemcitabine on NSCLC and pancreatic cancer are limited due to resistance [10,11]. Epidermal growth factor receptor (EGFR) signaling is activated in several types of solid cancer, including NSCLC, pancreatic cancer, and glioblastoma [12,13]. Thus, EGFR is a candidate target for molecular-targeted therapy for these types of cancers. However, the clinical benefits of first- and second-generation EGFR tyrosine kinase inhibitors (EGFR-TKIs) are limited in NSCLC, pancreatic cancer, and glioblastoma due to drug resistance [14,15,16,17,18,19,20,21]. For example, a phase III trial demonstrated that the combination of erlotinib, a first-generation EGFR-TKI, and gemcitabine improved the survival of patients with advanced pancreatic cancer; however, this improvement was limited [15]. In glioblastoma, clinical trials on EGFR-TKIs, such as gefitinib, erlotinib, and dacomitinib, demonstrated that their clinical benefits were limited or absent [18,19,20,21]. Osimertinib, a third-generation EGFR-TKI, was developed to overcome the EGFR-TKI-resistant T790M mutation in the EGFR gene that is frequently acquired during treatments with standard EGFR-TKIs for NSCLC [22,23]. Osimertinib is superior to standard EGFR-TKIs as first-line therapy for NSCLC [24]; however, resistance occurs, which limits its efficacy [25]. Furthermore, since the safety profile of osimertinib was found to be better than those of other standard EGFR-TKIs in clinical trials [23,24], it is one of the promising drugs for the treatment of pancreatic cancer and glioblastoma. Strategies that augment the effects of anticancer drugs, such as gemcitabine and osimertinib, are needed to improve the prognosis of these malignancies.

Drug repositioning or repurposing is a method that applies already-approved drugs to new indications [26,27]. The development of new drugs de novo is not only time-consuming and expensive, the success rate for drug approval is low [28]. All phases of clinical trials have been completed for already-approved drugs, and their safety profiles in humans are known. Thus, drug repositioning saves the time and costs associated with drug development and is attracting increasing interest by academia and industries. Drug repositioning has been successfully applied to several drugs. For example, finasteridine and minoxidil, which were originally developed to treat benign prostatic hyperplasia and hypertension, respectively, are now utilized for androgenic alopecia [29]. In the field of cancer therapy, non-oncological drugs are now being applied to the treatment of cancer: thalidomide, an anti-emetic drug, is now applied to the treatment of multiple myeloma [30], and everolimus, originally developed as an immunosuppressant, is used in the treatment of breast cancer, renal cell carcinoma, and neuroendocrine tumors [31,32,33,34].

Spironolactone, a classic potassium-sparing diuretic drug, is used to treat ascites in patients with cancer [35], hypertension derived from an adverse effect of anti-angiogenesis drugs [36,37], and brain edema caused by brain tumors [38]. Thus, spironolactone is a well-tolerated drug even in patients with cancer. Recent studies reported that spironolactone exerts anticancer effects by suppressing DNA damage repair and acts as a chemosensitizer in combination with DNA-damaging reagents, such as cisplatin [39,40,41]. Therefore, spironolactone is a good candidate drug for cancer therapy from the perspective of drug repositioning. However, it currently remains unclear whether spironolactone acts as a chemosensitizer of anticancer reagents that do not involve DNA damaging. In the present study, we examined the anticancer effects of spironolactone combined with gemcitabine and osimertinib, which are not DNA-damaging reagents, in non-stem cancer cells and the CSCs of NSCLC, pancreatic cancer, and glioblastoma in vitro and in vivo and investigated the mechanisms responsible for these anticancer effects.

## 2. Results

### 2.1. Spironolactone Inhibits Cancer Cell Growth and Is Cytotoxic to Cancer Cells, including CSCs, but not to Normal Cells

To examine whether spironolactone exerts cancer cell growth inhibitory effects, three representative cancer cell lines (A549, PANC-1, and PC-9) and one subline (PC-9-OR; osimertinib-resistant PC-9) were treated with spironolactone, and then subjected to cell viability assays. Spironolactone induced cell death and inhibited growth in these cell lines (Figure 1a and Appendix A). We then examined whether spironolactone exerts inhibitory effects on CSC lines. We treated two representative CSCs (A549 CSLC and PANC-1 CSLC cells) with spironolactone and then subjected them to cell viability assays. Spironolactone induced cell death and inhibited growth in CSCs (Figure 1b). We also examined the toxicity of spironolactone in non-cancer cells. We treated non-cancer cells (normal human fibroblasts [IMR-90]) with spironolactone, and then subjected them to cell viability assays. Spironolactone was not toxic to normal cells at the concentrations examined (Figure 1c). These results imply that spironolactone is not toxic to normal cells, but exerts cancer cell- and CSC-specific cytotoxic and growth inhibitory effects.

### 2.2. Spironolactone Decreases Resistance to Gemcitabine and Osimertinib

To examine whether spironolactone reduces resistance to anticancer reagents, A549, PANC-1, PC-9, and PC-9-OR cell lines were co-treated with spironolactone and an anticancer reagent, gemcitabine or osimertinib, and these treated cells were then subjected to cell viability assays. The results obtained showed that spironolactone reversed resistance to these anticancer reagents (Figure 2).

### 2.3. Spironolactone Reduces Resistance to Gemcitabine and Osimertinib in CSCs.

To examine whether spironolactone decreases resistance to anticancer reagents in CSCs (A549 CSLC and PANC-1 CSLC cells), which are highly resistant to cytotoxic reagents [42], these cells were co-treated with spironolactone and gemcitabine or osimertinib, and were then subjected to the viability assay. The results obtained showed that spironolactone decreased resistance to these anticancer reagents in CSCs (Figure 3).

### 2.4. Reductions in Survivin Expression Are Involved in the Potency of Spironolactone to Decrease Resistance to Gemcitabine and Osimertinib

We investigated the mechanisms by which spironolactone sensitizes cancer cells to gemcitabine and osimertinib. We previously reported that survivin, an anti-apoptotic protein, is involved in osimertinib resistance in glioma stem cells (GSCs), the CSCs of glioblastoma [43], and in NSCLC cells [44], and also that aripiprazole and brexpiprazole, antipsychotic drugs, reduce the expression of survivin and chemosensitize NSCLC and pancreatic cancer cells to gemcitabine [45,46]. Thus, cells treated with spironolactone were subjected to an immunoblot analysis of survivin, and the results obtained showed that survivin expression was decreased by the spironolactone treatment (Figure 4a,b). Survivin expression levels were markedly lower in IMR-90 cells than in A549 cells, and spironolactone only slightly reduced survivin expression levels (Appendix A). Low survivin levels may explain the weak responsiveness of IMR-90 cells to spironolactone. Furthermore, to elucidate the mechanisms underlying spironolactone-induced reductions in survivin levels, we examined changes in mRNA levels by spironolactone and alterations in protein levels by a treatment with MG132, a proteasome inhibitor. Spironolactone slightly decreased the mRNA levels of survivin (Appendix A), and the treatment with MG132 partially restored the expression of survivin (Appendix A). These results suggest that alterations in mRNA expression levels and proteasome degradation are at least partly involved in the regulation of survivin levels by spironolactone. To examine the involvement of survivin expression in resistance to gemcitabine and to confirm its involvement in osimertinib resistance, we treated A549 cells with YM155, a suppressor of survivin, or siRNA against survivin (siSurvivin) in combination with gemcitabine or osimertinib, and cells were then subjected to immunoblotting and cell viability assays (Figure 4c,d). The reduction of survivin by treatment with YM155 and siSurvivin sensitized the cells to gemcitabine and osimertinib, indicating that inhibition of survivin expression is sufficient to chemosensitize A549 NSCLC cells to gemcitabine and osimertinib. Together, these results suggest that reduction of survivin expression by spironolactone sensitizes these cells to gemcitabine and osimertinib most likely through suppression of survivin.

### 2.5. Spironolactone Exerts Anticancer, Chemosensitizing, and EGFR-TKI Sensitizing Effects with Survivin Reductions in Glioma Stem Cells

Since we previously reported that some treatment strategies for GSCs are mediated by survivin reductions [43,46], we examined whether spironolactone is a potential treatment candidate for GSCs. The results obtained showed that in GS-Y01, a patient-derived GSC line, spironolactone reduced cell viability and increased cell death (Figure 5a) and also reduced the expression of survivin (Figure 5b). Similar to previous findings showing that survivin is involved in the resistance of GSCs to osimertinib [43], the spironolactone treatment sensitized GSCs to osimertinib (Figure 5c).

### 2.6. Spironolactone Augments Anti-Tumor Effects of Osimertinib In Vivo

The results obtained thus far implied that spironolactone effectively overcomes the resistance of several types of cancer cells to osimertinib in vitro. To evaluate the therapeutic relevance of these results in vivo, we examined the efficacy of the systemic administration of spironolactone and osimertinib alone or in combination against subcutaneous xenografts of lung cancer cells. Since a pilot toxicity study showed that mice tolerated the combination of osimertinib 5 mg/kg (orally) and spironolactone 25 mg/kg (intraperitoneally) given 5 times a week, we used this treatment protocol and treated tumors formed by the subcutaneous implantation of A549 cells (EGFR-wt, intrinsically resistant to osimertinib) into nude mice. While neither osimertinib nor spironolactone given alone exerted appreciable effects over the control treatment, their combination inhibited the growth of tumors significantly more than the control treatment (Figure 6a). Furthermore, this combination did not appear to exert adverse effects, as assessed by body weight (Figure 6b). These results suggest that spironolactone successfully sensitized otherwise resistant lung cancer cells to osimertinib in vivo.

## 3. Discussion

Since cancer cells are resistant to many types of anticancer reagents by various mechanisms, extensive research has been performed to overcome chemoresistance. We examined the anticancer effects of spironolactone, a classic potassium-sparing diuretic that is used for cancer patients with ascites, hypertension induced by anti-VEGF therapy, and brain edema related to brain tumors, from the perspective of drug repositioning or repurposing. We revealed that spironolactone suppressed the expression of survivin, an anti-apoptotic protein, at a concentration that was harmless to non-cancer cells and also chemosensitized cancer cells and CSCs to anticancer reagents, such as gemcitabine and osimertinib. Therefore, spironolactone is a good candidate drug for the treatment of cancer.

There are several sources of chemoresistance to cancer including alterations to drug transporters, the suppression of apoptosis, mitochondrial alterations, the promotion of DNA damage repair, autophagy, epithelial-mesenchymal transition, and CSCs [2,3,4,5]. Previous studies reported that spironolactone inhibits DNA repair and chemosensitizes cancer cells to DNA-damaging reagents, such as cisplatin [39,40,41]. In the present study, we revealed that spironolactone chemosensitized cancer cells to gemcitabine and osimertinib. Gemcitabine is a nucleoside analog that is incorporated into DNA and prevents elongation of the DNA chain, leading to cell death [47,48]. Osimertinib is an inhibitor of epidermal growth factor receptor tyrosine kinase (EGFR-TKI) that suppresses EGFR signaling [24]. Since the main mechanism of action of gemcitabine and osimertinib is not damage to DNA, the chemosensitizing effects of spironolactone to gemcitabine and osimertinib may be mediated by mechanisms other than the suppression of DNA repair. We and others reported that drugs that suppress survivin chemosensitize cancer cells to EGFR-TKIs [43,48,49,50,51] and gemcitabine [52]. Survivin is a member of the inhibitor of apoptotic protein (IAPs) family, which suppresses apoptosis by modulating the execution of extrinsic and intrinsic apoptotic signals. We showed that spironolactone reduced the expression of survivin in cancer cells and sensitized these cells to these anticancer reagents. Similarly, the suppression of survivin by YM155, a transcriptional suppressor of survivin, or siRNA against survivin chemosensitized these cells to anticancer reagents. These results suggest that survivin is involved in chemoresistance to gemcitabine and osimertinib in cancer cells and the CSCs of NSCLC and pancreatic cancer, and also that spironolactone chemosensitizes NSCLC, pancreatic cancer, and glioblastoma cells to gemcitabine and osimertinib most likely through the suppression of survivin.

Survivin expression is regulated by several mechanisms, including mRNA expression and proteasome degradation. YM155, a clinically examined survivin suppressor, reduces survivin expression levels by suppressing the transcription of survivin. In the present study, we revealed that spironolactone reduced survivin mRNA levels to a lesser extent, and MG132, a proteasome inhibitor, partially restored spironolactone-induced decreases in survivin. These results suggest that spironolactone reduces survivin by mRNA reductions and increased degradation by proteasomes. Spironolactone induces the degradation of XPB, a major component of transcription factor II, and impairs transcription activity [40]. Furthermore, the ubiquitin-proteasome pathway regulates the stability of survivin [43,53,54,55]. The ubiquitination of survivin is controlled by the E3 ligases FBXL7 [53] and cullin 9 [54], and by the deubiquitinase USP9X [55]. Further studies are needed to elucidate the involvement of XPB and these ubiquitination-related enzymes in spironolactone-induced reductions in survivin.

Spironolactone is a well-known and widely-used potassium-sparing diuretic reagent that mainly antagonizes mineralocorticoid (or aldosterone) receptors. However, we noted that a representative mineralocorticoid receptor agonist, aldosterone, did not rescue the effects of spironolactone on survivin reductions (Appendix A). Moreover, more specific selective mineralocorticoid receptor antagonists, second- and third-generation potassium-sparing diuretic reagents, such as eplerenone [56] and esaxerenone [57], have more limited effects on survivin expression than the first-generation, less selective potassium-sparing diuretic, spironolactone (Appendix A). Similar to the present results, eplerenone was previously shown to not inhibit nucleotide excision repair (NER) or the proliferation of cancer cells [39,40]. Although the main target of spironolactone is mineralocorticoid receptors, spironolactone also binds to receptors related to mineralocorticoid receptors and exerts antiandrogen and weak progesterone properties as well as some indirect estrogenic and glucocorticoid effects [57]. These results imply the involvement of receptors other than mineralocorticoid receptors, such as glucocorticoid receptors, in chemosensitizing effects.

There are obstacles to applying the present results to clinical settings. The concentration used in our in vitro studies (25 µM) may be higher than the expected tissue concentration observed in humans. In other studies, similar concentrations of spironolactone (10 or 40 µM) were required to exert anti-cancer effects in vitro [39,40,41]. Since spironolactone is converted into several active metabolites (canrenone, 7α-thiomethyl-spironolactone, and 6β-hydroxy-7α-spironolactone) mainly in the liver in vivo [58] and shows complex tissue distribution, higher concentrations of spironolactone may be needed to exert anti-cancer effects in vitro. Further studies are needed to address this issue, as indicated previously by Gold et al. [39]. By taking dose conversions from mice to humans [59] into consideration, the dosage used in the in vivo study (25 mg/kg body weight) corresponds to 2.03 mg/kg for humans, which is an acceptable dosage used in clinical settings. The acceptable dosage of spironolactone showed combinational effects with osimertinib on tumor growth, and these results strongly suggest the clinical application of spironolactone for patients with cancer.

In conclusion, we herein reported that spironolactone exerted anticancer and chemosensitizing effects on various types of cancers most likely through survivin reductions. Importantly, the suppression of survivin is a novel mechanism for the chemosensitizing effects of spironolactone in combination with non-DNA-damaging chemotherapeutic reagents, such as gemcitabine and osimertinib. Therefore, spironolactone has potential as a promising anticancer drug for NSCLC, pancreatic cancer, and glioblastoma in combination with non-DNA-damaging anti-cancer drugs as well as DNA-damaging drugs with a known excellent safety profile.

## 4. Materials and Methods

### 4.1. Antibodies and Reagents

An anti-survivin (#2808) antibody was purchased from Cell Signaling Technology, Inc. (Beverly, MA, USA). An anti-β-actin (A1978) antibody was from Sigma (St. Louis, MO, USA). Gemcitabine was also from Sigma, and was dissolved in dimethyl sulfoxide (DMSO) to prepare a 1 mM stock solution. Spironolactone and eplerenone were from Tokyo Kasei Kogyo (Tokyo, Japan) and dissolved in DMSO to prepare a 100 mM stock solution. Osimertinib and YM155 were purchased from Chemscene LLC. (Monmouth Junction, NJ, USA) and dissolved in DMSO to 10 mM and 20 μM, respectively, as stock solutions. Aldosterone was from Toronto Research (North York, Canada) and dissolved in DMSO to a 100 mM stock solution. Esaxerenone was from Medchem (Monmouth Junction, NJ, USA) and dissolved in DMSO to a 50 mM stock solution. MG132 was from Wako Pure Chemical Industries (Osaka, Japan) and dissolved in DMSO to a 10 mM stock solution.

### 4.2. Cell Culture

The human non-small cell lung cancer (NSCLC) cell lines A549 and PC-9 were obtained from the Riken BioResource Center (Tsukuba, Japan). The human pancreatic cell line PANC-1 was from the Cell Resource Center for Biomedical Research, Institute of Development, Aging and Cancer, Tohoku University (Sendai, Japan). A549 and PANC-1 cell lines were both cultured in DMEM/F12 medium and PC-9 was cultured in RPMI1640 medium instead of DMEM/F12 medium, and these cell line media were supplemented with 10% fetal bovine serum (FBS), 100 units/mL of penicillin, and 100 μg/mL of streptomycin. An osimertinib-resistant subline of PC-9 (PC-9-OR) was established by culturing in increasing concentrations of osimertinib (0.1–1.5 μM) over a two-month period. PC-9-OR cells were maintained in the presence of 1.5 μM osimertinib. The establishment of A549 CSLC and PANC-1 CSLC was previously reported [60,61]. The authenticity of A549 CSLC and PANC-1 CSLC was verified by the genotyping of short tandem repeat (STR) loci (Bio-Synthesis, Inc., Lewisville, TX, USA) and comparisons with the ATCC STR database for Human Cell Lines. GS-Y01 is a CSC line derived from a glioma patient [62]. These CSCs were cultured as previously described [60,63,64]. Briefly, these cells were cultured on collagen I-coated dishes (IWAKI, Tokyo, Japan) in the stem cell culture medium (DMEM/F12 medium with 1% B27 supplement (Gibco-BRL, Carlsbad, CA, USA), 20 ng/mL of EGF and FGF2 (Peprotech, Inc., Rocky Hill, NJ, USA), D-(+)-glucose (final concentration, 26.2 mM), L-glutamine (final concentration, 4.5 mM), 100 units/mL of penicillin, and 100 μg/mL of streptomycin). Stem cell culture medium was changed every three days, and EGF and FGF2 were supplemented into the culture medium every day. IMR90, normal human fetal lung fibroblasts, were purchased from the American Type Culture Collection, and cultured in DMEM/F12 supplemented with 10% FBS, 100 units/mL penicillin, and 100 μg/mL streptomycin. All experiments with IMR90 were performed within a low passage number (less than eight).

### 4.3. Cell Viability Assays

Viable and dead cells were identified by their ability and inability to exclude vital dyes, respectively [52,64]. Briefly, cells were stained with 0.2% trypan blue, and the numbers of viable and dead cells were counted using a hemocytometer. Dead cells (%) were defined as 100 × ‘the number of dead cells’/(‘the number of viable cells’ + ‘the number of dead cells’). The following formula was used to obtain IC_50_ values, as previously reported [65]: IC_50_ = 10^[log(A/B) × (50-C)]/[(D-C) + Log(B)]^, where A and B are the corresponding concentrations of the test drug directly above and below 50% inhibition, respectively, and C and D are the percentages of inhibition directly below and above 50% inhibition, respectively.

### 4.4. Gene Silencing with siRNA

Two siRNAs against human survivin (BIRC5 #2; HSS 179404, #3; HSS 179405) and Medium GC Duplex #2 of Stealth RNAi^TM^ siRNA Negative Control Duplexes (non-targeting control) were obtained from Thermo Fisher Scientific (Waltham, MA, USA). Cells were transiently transfected with RNAs using Lipofectamine RNAiMAX^TM^ (Thermo Fisher Scientific) according to the manufacturer’s instructions.

### 4.5. Mouse Study

A mouse xenograft study was performed as previously described [52]. Regarding subcutaneous implantation, A549 cells (1 × 10^6^ cells) suspended in 200 µL of PBS were implanted subcutaneously into the flank region of 7-week-old male BALB/cAJcl-nu/nu mice (CLEA Japan, Inc., Tokyo, Japan) after being anesthetized with an intraperitoneal injection of medetomidine, midazolam, and butorphanol (0.3, 4, and 5 mg per kg body weight, respectively). After implantation, the general health status and presence of subcutaneous tumors were measured. Tumor volumes were assessed by measuring tumor diameters using a caliper and calculated as the larger diameter × the smaller diameter × height. Regarding the systemic administration of drugs, stock solutions of spironolactone (10 mg/mL) and osimertinib (2 mg/mL) were diluted in DMSO to prepare 100 µL solutions for each injection. Spironolactone was intraperitoneally administered to mice at 25 mg/kg body weight 5 times a week and osimertinib was orally administered at 5 mg/kg 5 times a week. The drug treatment was started after the confirmation of subcutaneous tumor formation, and tumor-bearing mice were randomized into 4 groups before the initiation of drug treatment. All animal experimental protocols were approved by the Animal Research Committee of Yamagata University (protocol code: 30027).

### 4.6. Immunoblot Analysis

Cells were washed with PBS and lysed in RIPA buffer (10 mM Tris-HCl [pH 7.4], 0.1% SDS, 0.1% sodium deoxycholate, 1% NP-40, 150 mM NaCl, 1 mM EDTA, 1.5 mM Na_3_VO_4_, 10 mM NaF, 10 mM sodium pyrophosphate, 10 mM sodium β-glycerophosphate, and 1% protease inhibitor cocktail set III Sigma). After centrifugation at 14,000× *g* at 4 °C for 10 min, supernatants were harvested as cell lysate samples, and the protein concentrations of the cell lysates were measured using a BCA protein assay kit (Pierce Biotechnology, Inc., Rockford, IL, USA). Cell lysates containing equal amounts of protein were separated by SDS-PAGE and transferred to polyvinylidene difluoride membranes. Membranes were probed with primary antibodies and then with an appropriate HRP-conjugated secondary antibody according to the manufacturer’s protocol. Immunoreactive bands were visualized using Immobilon Western Chemiluminescent HRP Substrate (Merck Millipore, Darmstadt, Germany). The relative density of immunoreactive bands was analyzed by densitometry using ImageJ 1.52a software (National Institutes of Health, Bethesda, MD, USA).

### 4.7. RNA Extraction and Reverse Transcription-PCR

Total RNA was isolated using Trizol (Thermo Fisher Scientific) according to the manufacturer’s instructions. RNA was reverse-transcribed into cDNA using the PrimeScript 1st strand cDNA Synthesis Kit (Takara Bio, Kusatsu, Japan). Synthesized cDNA samples were amplified by PCR using Quick Taq HS DyeMix (Toyobo, Osaka, Japan). The primer sequences used were as follows: actin (5′-CCCATGCCATCCTGCGTCTG-3′ [forward] and 5′-CGTCATACTCCTGCTTGCTG -3′[reverse]), and survivin (5′-CCTTTCTCAAGGACCACCGCATCT-3′ [forward] and 5′-CGCACTTTCTCCGCAGTTTCCT-3′ [reverse]).

### 4.8. Statistical Analysis

Results are expressed as means and standard deviations (SD) and differences were compared using the two-tailed Student’s *t*-test. *p*-values < 0.05 were considered to be significant and were indicated with asterisks.

## 5. Conclusions

The anticancer and chemosensitizing effects of spironolactone on various types of cancer cells appear to be attributed to survivin reductions, and the suppression of survivin is a novel mechanism for the chemosensitizing effects of spironolactone in combination with non-DNA-damaging chemotherapeutic drugs, such as gemcitabine and osimertinib.

## Figures and Tables

**Figure 1 cancers-11-01550-f001:**
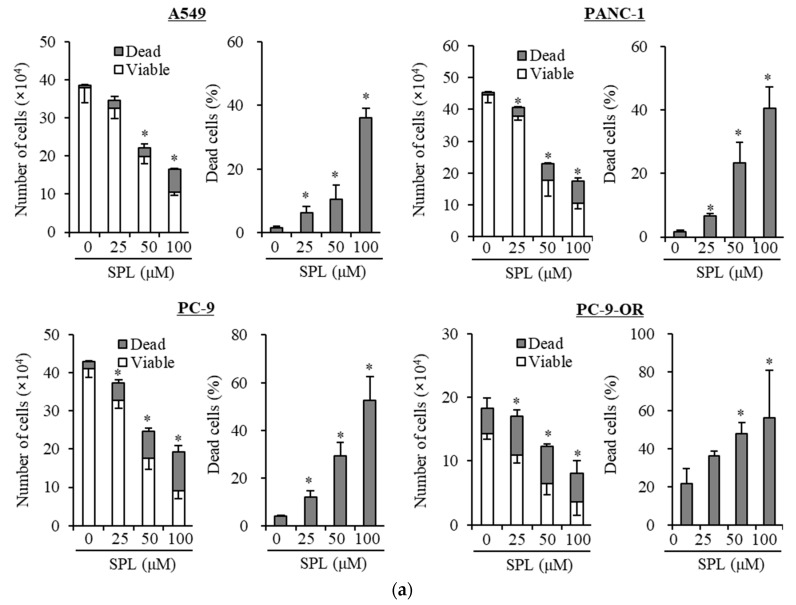
Spironolactone selectively induces cell death and inhibits the growth of cancer cells. A549, PANC-1, PC-9, and PC-9-OR (osimertinib-resistant) (**a**), cancer stem cell lines (A549 CSLC and PANC-1 CSLC) (**b**), and IMR90 normal human fibroblasts (**c**) were treated without (as control) or with the indicated concentrations of spironolactone (SPL) for three days, and the numbers of viable and dead cells (left panels) as well as the percentage of dead cells (right panels) were then assessed. Values in the graphs represent the means ± SD of triplicate samples of a representative experiment repeated with similar results. * *p* < 0.05.

**Figure 2 cancers-11-01550-f002:**
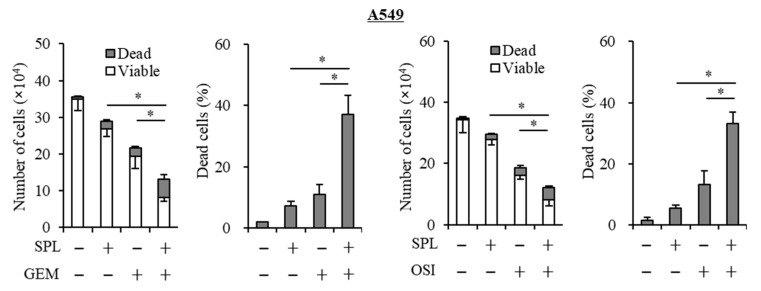
Spironolactone sensitizes cancer cells to anticancer reagents. Cells were treated with the indicated chemotherapeutic reagents (GEM, gemcitabine, 0.1 µM; OSI, osimertinib, 2 µM) in the absence or presence of 25 µM spironolactone (SPL) for three days, and the numbers of viable and dead cells (left panels) as well as the percentage of dead cells (right panels) were then assessed. Values represent the means ± SD of triplicate samples of a representative experiment repeated with similar results. * *p* < 0.05.

**Figure 3 cancers-11-01550-f003:**
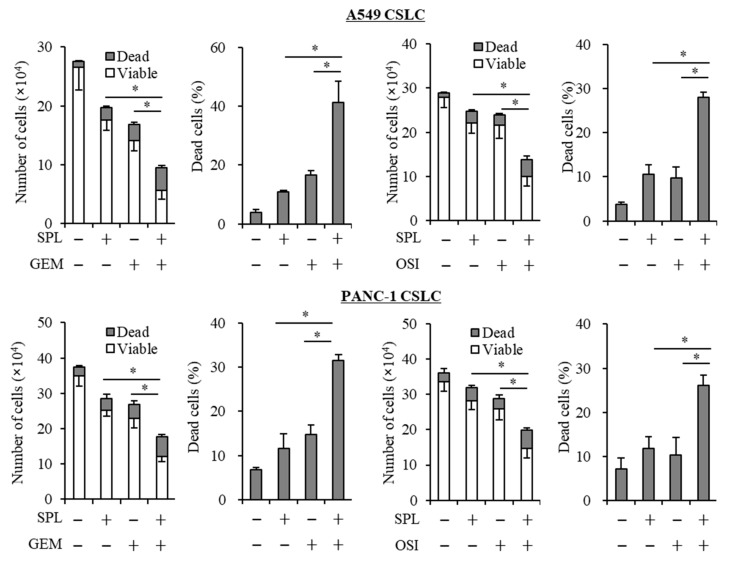
Spironolactone sensitizes cancer stem cells to chemotherapeutic reagents. Cancer stem cells were treated with the indicated chemotherapeutic reagents (GEM, gemcitabine, 0.25 µM; OSI, osimertinib, 2 µM) in the absence or presence of 25 µM spironolactone (SPL) for three days, and the numbers of viable and dead cells (left panels) as well as the percentage of dead cells (right panels) were then assessed. Values represent the means ± SD of triplicate samples of a representative experiment repeated with similar results. * *p* < 0.05.

**Figure 4 cancers-11-01550-f004:**
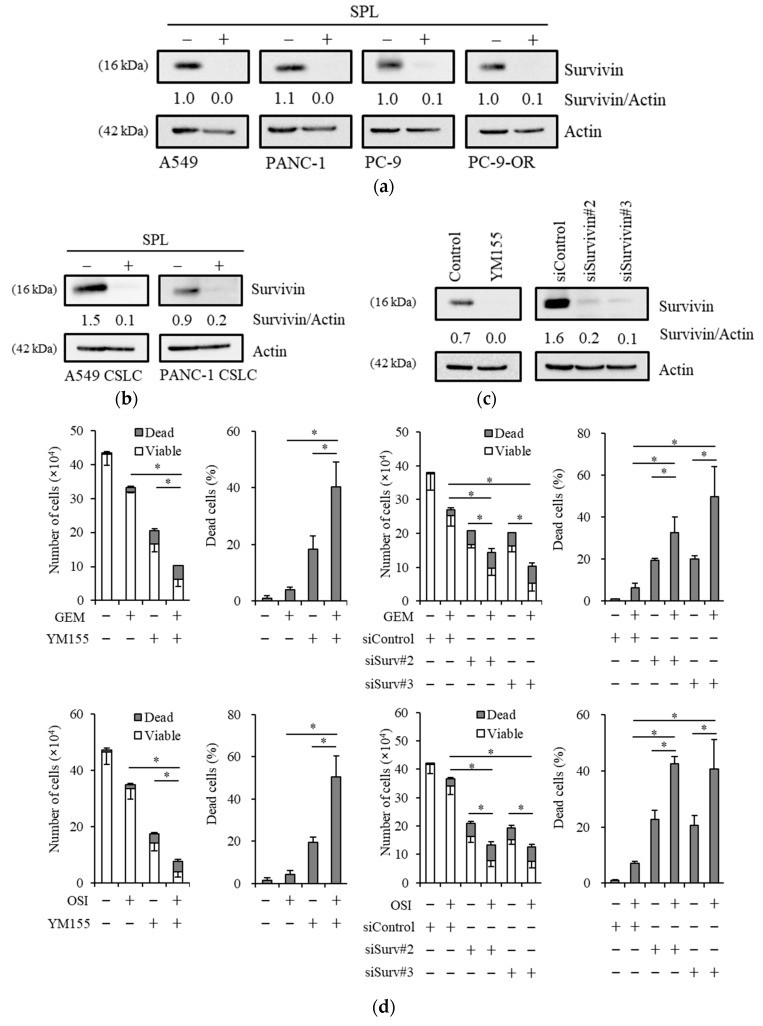
Spironolactone decreases survivin expression and survivin reductions imitate the chemosensitizing effects of spironolactone. Cells treated with or without 25 µM spironolactone (SPL) for three days were subjected to an immunoblot analysis for survivin expression (**a**,**b**). YM155, a pharmacological survivin inhibitor, at the concentration of 10 nM and the knockdown of survivin reduced survivin expression in cancer cells (A549) (**c**). The pharmacological or genetic inhibition of survivin sensitized cancer cells (A549) to chemotherapeutic reagents (GEM, gemcitabine, 0.1 µM; OSI, osimertinib, 2 µM) (**d**). * *p* < 0.05.

**Figure 5 cancers-11-01550-f005:**
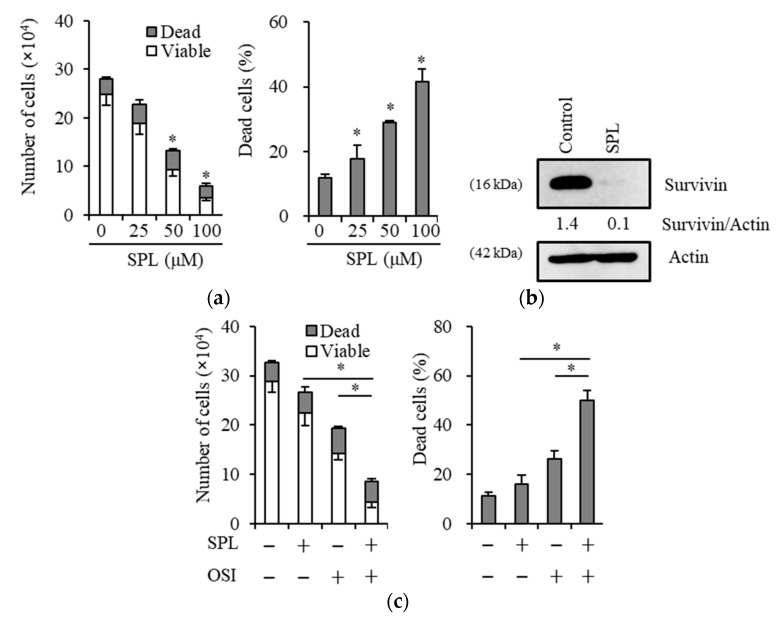
Spironolactone induces similar effects in GS-Y01, a glioma stem cell line, to those in cancer stem cells (CSCs) of Non-small cell lung cancer (NSCLC) and pancreatic cancer. GS-Y01 cells were treated without (as control) or with the indicated concentrations of spironolactone (SPL) for three days, and the numbers of viable and dead cells (left panel) as well as the percentage of dead cells (right panel) were then assessed (**a**). Cells treated with or without spironolactone (SPL) for three days were subjected to an immunoblot analysis for survivin expression (**b**). Glioma stem cells were treated with the indicated chemotherapeutic reagents (OSI, osimertinib) in the absence or presence of spironolactone (SPL) for three days, and the numbers of viable and dead cells (left panel) as well as the percentage of dead cells (right panel) were then assessed (**c**). Values in the graphs represent the means ± SD of triplicate samples of a representative experiment repeated with similar results. * *p* < 0.05.

**Figure 6 cancers-11-01550-f006:**
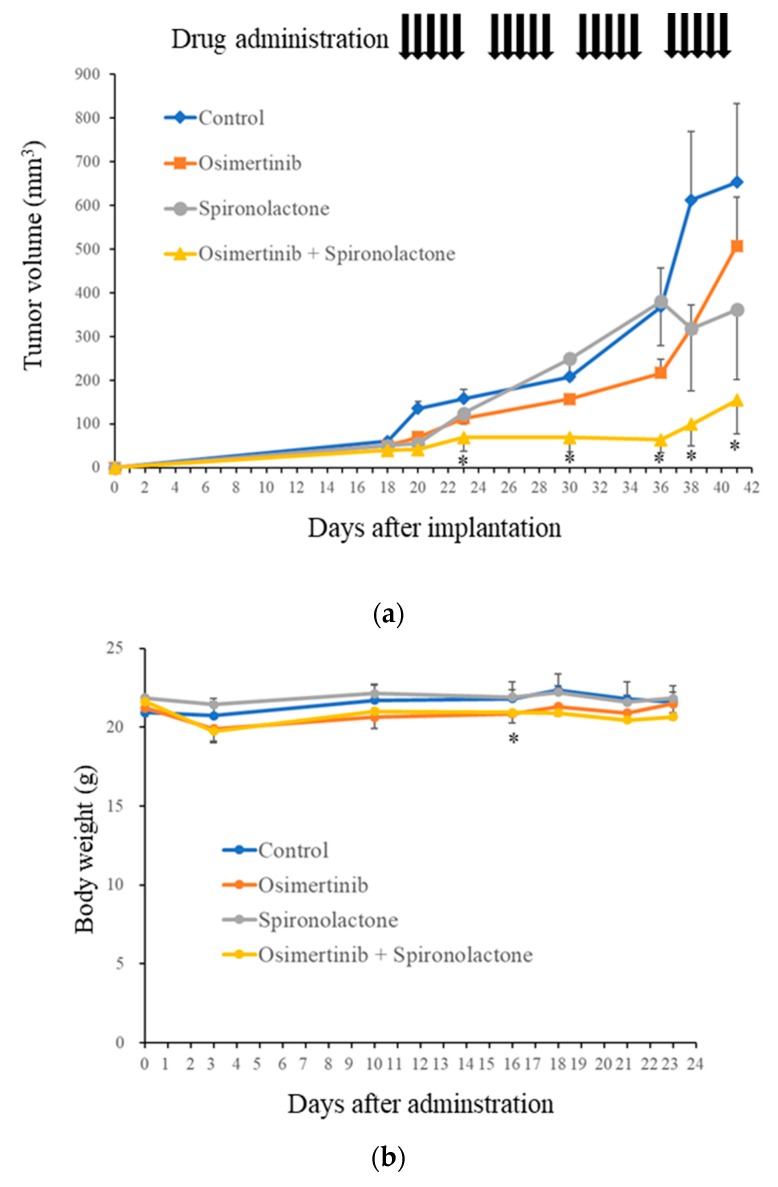
Spironolactone sensitizes cancer cells to osimertinib in vivo. Mice (five for each group) were subcutaneously implanted with A549 cells. After the confirmation of tumor formation, mice were treated with or without osimertinib and/or spironolactone as detailed in the Materials and Methods. Tumor volume (**a**) and mouse body weight (**b**) were measured, and the results obtained are presented in the graphs as the means ± SD of each group. * *p* < 0.05, compared with control group in (**a**) and comparison between spironolactone treatment group and combination group in (**b**).

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
