# Peer review of "Spironolactone, a Classic Potassium-Sparing Diuretic, Reduces Survivin Expression and Chemosensitizes Cancer Cells to Non-DNA-Damaging Anticancer Drugs"

_cancers, 2019, doi:10.3390/cancers11101550_

Round 1
Reviewer 1 Report
In this paper, authors show that Spironolactone, a diuretic, can show anti-cancer effect on lung, pancreatic or glioblastoma cancer cells. They show that Spironolactone impacts Survivin expression, explaining the impact of Spironolactone on the augmentation of cell death by gemcitabine and osimertinib. Overall this paper probably deserve publication in Cancers Journal if the authors perform some complementary experiments described below.
1- The authors should attempt to deeper understand the molecular basis of Survivin degradation by Spironolactone. Is it due to a transcriptional silencing ? How does Survivin mRNA evolve after SP treatment ? Is it due to protein degradation? Is it reversible ? Does the proteasome involved ? SP induces the degradation of a crucial subunit of TFIIH, XPB. Since TFIIH is a basal transcription factor, does its degradation impact Survivin expression ?
2- Is survivin expressed in IMR-90 cells ? If yes, does spironolactone decreases its expression in IMR-90 cells ?
3- Authors should provide the IC50 of their cancer cells treated with gemcitabine and osimertinib with or without Spironolactone
Reviewer 2 Report
This is clearly written manuscript with clearly presented data. However I have a few question for authors and I suggest to perform one additional experiment to further confirm the significance of this study.
Fig. 1. How where the CSCs lines selected? Which antigen or other features were used for section?.
Did You compare the expression of most important drug transporters between basic and CSCs cell lines?
Are there any data if Spironolactone a substrate for drug transporters?
What is a stable therapeutic concentration of investigated drugs? Are the drug concentrations used in this study close to therapeutic concentration?
What did you used the trypan blue assay instead of MTT assay or other that measure cell metabolism?
Animal study. I suggest to use IHC method to compare the expression of Survivin in tumours between all treated groups or at least between control and Spironolactone treated tumours. This give the answer if the effect of Spironolactone is also related to changes in Survivin expression in vivo. If You have fixed tumours tissue from experiments You can use them in this study.
Round 2
Reviewer 1 Report
Authors answered to my major concerns. Paper can be published in its actual form.
Reviewer 2 Report
Dear Authors
Thank You for response.
Radosław Januchowski